# Autothermal Reforming of Volatile Organic Compounds to Hydrogen-Rich Gas

**DOI:** 10.3390/molecules28020752

**Published:** 2023-01-11

**Authors:** Chao Bian, Jiazhun Huang, Biqi Zhong, Zefeng Zheng, Dai Dang, Obiefuna C. Okafor, Yujia Liu, Tiejun Wang

**Affiliations:** 1School of Chemical Engineering and Light Industry, Guangdong University of Technology, Guangzhou 510006, China; 2Corning Incorporated, Corning, NY 14814, USA

**Keywords:** autothermal reforming, VOCs, hydrogen, nickel-based catalyst

## Abstract

Industrial emissions of volatile organic compounds are urgently addressed for their toxicity and carcinogenicity to humans. Developing efficient and eco-friendly reforming technology of volatile organic compounds is important but still a great challenge. A promising strategy is to generate hydrogen-rich gas for solid oxide fuel cells by autothermal reforming of VOCs. In this study, we found a more desirable commercial catalyst (NiO/K_2_O-γ-Al_2_O_3_) for the autothermal reforming of VOCs. The performance of autothermal reforming of toluene as a model compound over a NiO/K_2_O-γ-Al_2_O_3_ catalyst fitted well with the simulation results at the optimum operating conditions calculated based on a simulation using Aspen PlusV11.0 software. Furthermore, the axial temperature distribution of the catalyst bed was monitored during the reaction, which demonstrated that the reaction system was self-sustaining. Eventually, actual volatile organic compounds from the chemical factory (C_9_, C_10_, toluene, paraxylene, diesel, benzene, kerosene, raffinate oil) were completely reformed over NiO/K_2_O-γ-Al_2_O_3_. Reducing emissions of VOCs and generating hydrogen-rich gas as a fuel from the autothermal reforming of VOCs is a promising strategy.

## 1. Introduction

Recently, industrial emissions of volatile organic compounds (VOCs) have increased dramatically, representing a serious threat to the entire ecosystem [1,2]. VOCs are important precursors for the formation of secondary pollutants such as fine particulate matter (PM2.5) and ozone (O_3_), which in turn cause atmospheric environmental problems such as haze and photochemical smog [3]. The VOCs pollutants were mainly emitted by chemical and petrochemical sectors, via processes such as petroleum refining, paint manufacture, waste combustion, and vehicle manufacturing [4,5]. According to the prediction by some researchers, industrial VOCs emissions will reach 33.4, 61.2 and 135.5 × 10^9^ kg Carbon/year by 2020, 2030 and 2050, respectively [6,7].

The most common and toxic non-halogenated compounds in industry include formaldehyde, benzene, carbon monoxide, toluene, propylene, phenol, acetone and styrene [8]. Toluene is the most emitted type of VOC and is difficult to reform [9,10]. Therefore, toluene is usually used as a model compound for VOCs to study. Many researchers have focused their attention on the treatment of VOCs for the enormous emissions and harmfulness. The negative environmental impact, toxicity of VOCs, and the increasingly stringent regulations required the investigation of more effective VOCs reforming technologies [11,12].

Numerous techniques, including non-destructive and destructive, have been used for eliminating VOCs. The non-destructive methods were mainly adsorption, absorption, condensation and membrane separation, and the latter includes catalytic reforming, thermal combustion, nonthermal plasma, photocatalytic degradation, and biological treatment. However, each technology and method has its limitations in removal efficiency, energy consumption, by-products as well as safety [13]. A process or technology enabling complete reforming of VOCs at lower cost and higher efficiency is urgently needed.

At present, the process used in industry to treat with VOCs is direct-combustion after they go through a concentration process. Although this process is effective, it is expensive to operate, requires frequent maintenance and addition extra process. Interestingly, the emission of VOCs can be reformed hydrogen-rich gas as a fuel for SOFCs. Solid oxide cells can directly convert chemical energy from hydrocarbon fuels (such as natural gas, biogas, etc.) into electrical energy with a variety of feedstocks on account of its higher efficiency and superior temperature resistance [14]. The process is not limited by the Carnot cycle, and does not release harmful substances externally [15].

While current abatement strategies require additional fuel to heat the reactor, the proposed approach not only eliminates the use of additional fuel, but also allows the use of VOCs, which are considered waste, to generate electricity [16]. As a result, this technology can result in significant cost savings. However, compared with the direct-combustion process, catalytic reforming is a superior approach in terms of economic efficiency of VOC elimination [17], lower operation temperatures, reduced by-products (CO_2_ and H_2_O) and reduced energy consumption [18].

Methods include catalytic partial oxidation, steam reforming, autothermal reforming (ATR), and oxidative steam reforming (OSR) is also autothermal reforming. Steam reforming (SR) is a strongly endothermic reaction, which requires high operating temperatures (700–900 °C) to ensure a consistent conversion of reactants to H_2_ [19]. The process of SR is defined by the following overall reaction:CnHm + nH_2_O ↔ CO_X_ + H_2_ + H_2_O, ∆H > 0(1)

The formation of carbon compounds and its high energy requirement are the main disadvantages of steam reforming [20,21]. It is worth noting that excess water added inhibits the reaction and increases energy consumption. 

A catalytic partial oxidation (CPO) reaction is an exothermic reaction in which a moderate amount of oxygen promotes the decomposition of hydrocarbons to H_2_, CO_2_, and H_2_O according to the reaction in Equation (2). The disadvantage of CPO is that it produces a lower H_2_ yield due to hydrogen reacts with oxygen [22,23]. Partial oxidation proceeds via the following overall reaction Equation (2):CnHm + nO_2_ ↔ CO_X_ + H_2_ + H_2_O, ∆H < 0(2)

Almost all reforming processes are usually accompanied by a water-gas shift (WGS) reaction (3) [24]. For the WGS reaction, water generation has an impact on H_2_ production.
CO + H_2_O ↔ CO_2_ + H_2_, ∆H < 0(3)

The process of autothermal reforming (ATR) is a combination of water-gas shift reaction (WGSR), catalytic partial oxidation and steam reforming [25]. Autothermal reforming in Equation (4) can reduce energy consumption in external heating furnaces since the heat generated by the exothermal partial oxidation can supply energy to the endothermal steam reforming reaction [26,27]. The main advantages of the autothermal reforming include increased energy efficiency and process stability, as demonstrated by thermodynamic studies [28]. For different autothermal reforming reactions, the selection of proper molars of steam-to-carbon, oxygen-to-carbon and catalysts were particularly essential for different autothermal reforming reactions [29].
CnHm + nO_2_ + mH_2_O ↔ CO_X_ + H_2_ + H_2_O, ∆H ~ 0(4)
CO + 3H_2_ ↔ CH_4_ + H_2_O, ∆H < 0(5)

Catalysts are crucial factors for autothermal reforming of VOCs. Precious metal catalysts are primarily used in industrial applications at present, while the low content of precious metals in the earth’s crust and high prices limit the prospects for their use [30]. Hence, non-precious metal catalysts are of interest to researchers. Ideal catalysts for autothermal reforming of VOCs must have high activity, selectivity, and stability when complicated volatile organic compounds are being reformed [31,32]. Nickel-based catalysts have an excellent ability to break C-C, C-H, and C-O bonds compared to other non-precious metals [33,34]. They are also preferred because of their inherent catalytic activity, as well as their abundance [35,36]. However, nickel quickly loses its activity due to sintering and carbon deposition [37]. Interestingly, the combination of basic activity metal oxides and nickel could inhibit coke deposition, such as K_2_O and CaO. To improve the durability of Ni catalysts, suitable supports carriers were selected to form bimetallic or multi-metallic doped catalysts based on well-dispersed Ni catalysts such as ZrO_2_, TiO_2_, La_2_O_3_, Al_2_O_3_ [38]. Metal oxide supports had a great influence on the performance of catalysts and could improve the stability of catalysts and increase the mobility of oxygen (optimizing the selectivity toward total oxidation) through the interaction with the active species [39]. The γ-Al_2_O_3_ support is used extensively in the petrochemical industry due to its large availability in nature and its excellent thermal stability [40,41].

Inspired by the above reports, in this work, we propose a new approach to supply hydrogen-rich fuel gas using autothermal reforming process of VOCs. The operation conditions of autothermal reforming of toluene were optimized by thermodynamic analysis using Aspen PlusV11.0 software (AspenTech, Bedford, MA, USA). NiO/K_2_O-γ-Al_2_O_3_ materials were utilized as the autothermal reforming catalyst compared with three different Ni-based catalyst (NiO/γ-Al_2_O_3_, NiO/Al_2_CaO_4_, and NiO/K_2_O-γ-Al_2_O_3_). The results showed the excellent ability to produce hydrogen-rich gas from VOCs by ATR over the NiO/K_2_O-γ-Al_2_O_3_ catalyst. Subsequently, under the optimal operating conditions, actual volatile organic compounds (C_9_, C_10_, toluene, paraxylene, diesel, benzene, kerosene, and raffinate oil) were completely reformed to hydrogen-rich gas. The temperature distribution of catalyst bed showed that catalytic partial oxidation occurred at the top section of catalyst bed, and steam reforming occurred at the lower section of catalyst bed. The thermodynamic calculations and the temperature distribution of catalyst bed proved that the system was self-sustaining. A comparison of the experimental result with lower heating value (LHV) calculation, as well as the efficiency of autothermal reforming processes was evaluated.

## 2. Results and Discussion

### 2.1. Thermodynamic Analysis

The thermodynamic equilibrium compositions in the reactor were calculated using the Aspen PlusV11.0 software (the AspenTech in USA.) based on the Gibbs free energy minimization method (Rgibbs reactor and SRK equation) in order to improve the ability to predict gas densities. The SRK equation is one of the most popular cubic equations currently used in research, simulations and optimizations that vapor-liquid equilibrium (VLE) properties [42,43]. Using the Lagrange multiplier method, the Gibbs free energy minimization for the system can be represented by the equations (Equations (6)–(8)), considering each species in the gas phase and the total reaction system [44]. In this analysis, the formation of the following compounds was considered: H_2_, CO, CO_2_, CH_4_, H_2_O, toluene, and the solid carbon is pure solid carbon.
(6)$$∆G_{fi}^{{}^{\circ}g}+TRln\frac{y_{i}\varphi_{i}P}{P^{{}^{\circ}}}+\underset{k}{\sum }\alpha_{ik}\lambda_{k}=0$$
(7)$$\overset{N}{\underset{i=1}{\sum }}n_{i}\left(∆G_{fi}^{{}^{\circ}g}+TRln\frac{y_{i}\varphi_{i}P}{P^{{}^{\circ}}}+\underset{k}{\sum }\alpha_{ik}\lambda_{k}\right)+n_{c}∆G_{fc\left(s\right)}^{{}^{\circ}}=0$$
(8)$$\underset{i}{\sum }n_{i}a_{ik}=A_{k}$$
where $∆G_{fi}^{{}^{\circ}g}$ is the standard Gibbs free energy of formation of gaseous species *i*, $y_{i}$ is the mole fraction of species *i*, $\varphi_{i}$ is the fugacity coefficient of species *i*, *P* and *P*° are the system pressure and standard state pressure (1 atm), respectively, $\alpha_{ik}$ is the number of the atoms of the *k*th element present in each molecule of gaseous species *i*, $\lambda_{k}$ is the Lagrange multiplier, and *A_k_* is the total atomic mass of the *k*th element in the feed. $n_{i}$ and $n_{c}$ are the mole number of species *i* and the solid carbon in the system, respectively, while$\;∆G_{fi}^{{}^{\circ}g}$ is the standard Gibbs free energy of formation of the solid carbon which is assumed to be zero.

Compared to previous studies, this study focused on toluene autothermal reforming to investigate the effect of different operating temperatures, O_2_/C molar ratios, and S/C molar ratios on chemical and energy performance metrics. Then, the reforming processes were compared at fixed operating conditions to investigate energy-related aspects in-depth and to explore the potential of operating the VOCs reforming under autothermal conditions.

It was essential to notice that chemical reactions did not always reach equilibrium since they were not kinetically feasible under certain conditions. As a result, this research should be regarded as an indicative approach upon which the direction of catalyst optimization was found. In actuality, the equilibrium overlooked reaction kinetics, catalytic characteristics, and transport processes. The results of this thermodynamic analysis can be compared to more thorough experimental data in the next step.

The operating temperature had a greater influence on the reaction system among the oxygen-to-carbon and steam-to-carbon. As shown in Figure 1a, thermodynamic analysis of autothermal reforming was carried out in a wide temperature range of 500–900 °C to thoroughly investigate the effect of temperatures on equilibrium product distribution. With the increasing of the temperature, the fraction of hydrogen increased and then decreased slightly. Meanwhile, it was observed that the energy consumption for external heating increased significantly, and the most molar fraction of hydrogen was observed at 700 °C without carbon deposition.

In the autothermal reforming process, the reaction of toluene with oxygen was a rapid combustion reaction in the high temperature. The effect of O_2_/C on the ATR of hydrogen production was also significant. It was varied between 0 and 1 under atmospheric pressure by the simulation. As shown in Figure 1b, the O_2_/C was required to be maintained at a relatively low value to ensure high hydrogen production since oxygen reacts with the hydrogen product. The optimal operating condition for maximizing hydrogen production was O_2_/C = 0.225.

In addition to the effect of temperature and O_2_/C, the effect of S/C was also studied in this work. As shown in Figure 1c, it could be observed that the composition of hydrogen increases with the increase of S/C. However, a higher S/C would result in the energy consumption and expensive production costs, which are not conducive to industrial applications. Considering the foregoing, the optimal operating condition was that temperature be equal to 700 °C, the value of oxygen-to-carbon be equal to 0.225 and the value of steam-to-carbon be equal to 1.31.

### 2.2. Performance of ATR over Ni-Based Catalyst

To analyze the role of different components in the NiO/K_2_O-γ-Al_2_O_3_, three Ni-based catalysts were tested for autothermal reforming under the same conditions. According to Figure 2, the NiO/γ-Al_2_O_3_ catalyst exhibited a similar distribution of gas products to the NiO/Al_2_CaO_4_ catalyst, showing that the addition of calcium oxide had little effect on the activity and selectivity of catalyst. The catalyst containing calcium oxide produced slightly lower amount of hydrogen and CO than NiO/γ-Al_2_O_3_, indicating that the addition of calcium oxide can absorb CO_2_ and promote the reverse water-gas shift reaction. This phenomenon is consistent with that reported in the literature [45,46]. However, comparing NiO/γ-Al_2_O_3_ catalyst and NiO/Al_2_CaO_4_ catalyst, NiO/K_2_O-γ-Al_2_O_3_ catalyst produced higher CO_2_ yield and lower CO yield. As the same time, NiO/K_2_O-γ-Al_2_O_3_ produced higher hydrogen yield and lower CH_4_ yield. Experimental results suggested that the addition of potassium oxide was able to increase the alkalinity of the catalyst and inhibit the methanation reaction [47,48]. In addition, the results of the toluene reforming fitted well with the simulation results over NiO/K_2_O-γ-Al_2_O_3_ catalyst.

According to Table 1, based on the comparison of catalytic performances with reported catalysts for toluene steam reforming, autothermal reforming exhibited a relatively higher hydrogen yield and conversion of toluene. Autothermal reforming of toluene has relatively lower operating temperatures compared with that reported in Table 1. The toluene conversion of 100% was obtained at 700 °C and hydrogen yield was up to 78.99% in the NiO/K_2_O-γ-Al_2_O_3_.

### 2.3. Effect of Operating Conditions on ATR Performance

The thermodynamic equilibrium model proposed was validated by comparison of the simulation and experimental data under the different operational conditions. Adjusting the operating conditions allowed the experimental results to be closer to the equilibrium values. In the following paragraphs, the effect of different operating conditions on ATR was analyzed.

#### 2.3.1. Effect of Temperature

As shown in Figure 3a, the experimental and simulated results were varied with temperatures ranging from 550 °C to 700 °C, which showed a very good fit of the equilibrium with simulated data, and the reaction was close to equilibrium within the whole temperature range. Within these operating temperatures, the catalytic partial oxidation showed the increase in H_2_ and CO production. However, the decrease in CO_2_ may be due to the reverse water-gas shift reaction. The higher operating temperature would improve the exothermic reverse water-gas shift reaction and increase the yield of CO, which is exothermic. Meanwhile, the content of CH_4_ was decreased according to reaction Equation (5), this process was inhibited at high temperatures.

To confirm whether the reaction system was self-sustained, the actual reaction temperature of the catalyst bed was measured. As shown in Figure 3b, it was noticed that the inner-layer and outer-layer temperatures of the catalyst bed were inconsistent under the fixed external control of temperature. The reaction temperature inside the catalyst bed and the fraction of hydrogen were increased with the increase of the control temperature, indicating that the higher catalyst bed temperature was a benefit for catalytic partial oxidation of toluene. In addition, the top section of the catalyst temperature was higher than the lower section, which indicated that a strong catalytic partial oxidation reaction occurred at the top section of the catalyst bed. The heat released from the catalytic partial oxidation reaction could be transferred to the lower section with the direction of material transfer to promote the steam reforming reaction, and the steam reforming reaction reacted at the lower section in the catalyst bed. When the external control temperature was equal to 700 °C, the temperature of the catalyst bed was higher than 700 °C, this result proved that the system was self-sustaining.

#### 2.3.2. Effect of O_2_/C

As shown in Figure 4a, with the increase in O_2_/C in the feedstocks, the fraction of hydrogen was slightly decreased resulting in excess oxygen reacting with hydrogen. The experimental and simulated results for CO and CO_2_ followed the same trend but were slightly different. The fraction of CO in the experiment was lower than the simulated results. The fraction of CO_2_ in the experiment was higher than the simulated results. Since CO was converted to CO_2_ with the increase in O_2_/C due to the combustion reaction and the water-gas shift reaction. According to Figure 4b, it was observed that the reaction temperature in the catalyst bed was rapidly increasing with the increase in O_2_/C, which resulted in increased catalytic partial oxidation. The heat released from the CPO contributed to the steam reforming and reduced the energy consumption of the external heating system thereby improving energy utilization. Figure 4b showed that when O_2_/C > 0.25, the reaction temperature in the catalyst bed was higher than the external control temperature, indicating that the reactor achieved a self-sustaining reaction.

It was noted that the fraction of gas production was more sensitive to O_2_/C molar ratios compared to operating temperature and S/C molar ratios, because the reaction rate of CPO was higher the steam reforming. The coupling of a strong exothermic CPO reaction and weaker steam reforming reaction was the main reason for the increase in catalyst bed temperature. The reaction of autothermal reforming is also accompanied by axial temperature diffusion, and most of the heat generated by the oxidation reaction is consumed at higher O*_2_*/C conditions and is not fully supplied to the steam reforming.

The exothermic oxidation reaction caused by toluene and oxygen can cause an uncontrollable temperature increase result in sintering of the catalyst and losing reaction activity because the rate of toluene in partial oxidation reforming is faster than steam reforming. A higher O*_2_*/C ratio will enhance the non-uniform temperature distribution and the temperature of the catalyst bed will increase significantly. A reasonable value of O*_2_*/C can be coupled well CPO reaction with the steam reforming reaction. The non-uniform axial temperature distribution in the catalyst bed can produce unstable hydrogen yield and decrease the durability of catalyst.

#### 2.3.3. Effect of S/C

The experimental results agreed with the simulated results and followed the same trend for values of S/C from 1.0 to 3.0. Figure 5a showed that with the increase in S/C, the fraction of H_2_ and CO increases slightly, while the fraction of CO_2_ decreases. The variations were due to the enhancement of the steam reforming reaction as a result of the increase of hydrogen production. Meanwhile, the addition of water promotes the water-gas shift reaction. However, the increase in H_2_ production promoted the production of methane and part of hydrogen is consumed according to the Le Chatelier principle, as shown in Equation (5).

As shown in Figure 5b, the upper catalyst temperature was higher than the lower layer under low S/C operating conditions. After the reactants entered the reactor and vaporized on the upper catalyst in the surface, catalytic partial oxidation reaction and steam reforming were occurring, accompanied by a water-gas shift reaction in the reaction system. Catalytic partial oxidation reaction dominated since it was a quick reaction compared with steam reforming. As the heat released by catalytic partial oxidation from the upper part of the catalyst bed was transferred downward to the lower part of the catalyst bed, it led to the increase in temperature at the lower catalyst layer and promoted steam reforming. With the increase in S/C, the catalytic partial oxidation reaction was suppressed and the steam reforming reaction was enhanced by absorbing the heat. When the steam-to-carbon value was equal to 2.4, the out-layer temperature in the catalyst bed was close to the external control temperature, indicating the reactor achieved a self-sustaining reaction.

The fixed bed reactor has a number of unusual characteristics including the flowing continuity and immovable the catalyst particles to the intensity of the heat transfer and nearly isothermal conditions throughout the reactor. However, a fixed bed reactor involves some complex flows and reaction such as CPO reaction and steam reforming reaction. Here, the experimental compositions measured at the top and the bottom of this catalyst bed reactor for a wide range of exit temperatures (Texit = 680–820 °C) are not uniform distribution of temperature. More uniform distribution along the length of the reactor compared to the top catalyst bed. Since the outlet in the flow direction was located at the lower end of the reactor, heat transfer was also present along the radial length of the reactor. The heat transfer coefficient in turn decreases gradually along the axial direction of the reactor. In the lower and middle portions of the catalyst bed, the reactor will combine exothermic and heated chemical reactions for thermal management purposes without the aid of external energy sinks or sources. By directly coupling both steam reforming and oxidation reactions within the same reactor catalyst bed, thereby eliminating the heat transfer bottleneck that is a feature of many chemical manufacturing, the direct coupled self-heating operation can lead to significant process improvements, improvements and cost savings that can be realized.

#### 2.3.4. ATR of Real VOCs

In this portion of the work, different VOCs from the Sinopec Shanghai Petrochemical Company Limited were reformed at optimal operating conditions, and the results were shown in Figure 6. According to Table 2, the raffinate oil had the highest component of hydrogen, and no other alkanes were produced at ATR [54]. The high carbon composition of the large hydrocarbons of C_9_, C_10_, and diesel produced some alkanes, mainly ethylene, indicating that long-chain alkanes had difficulty in autothermal reforming [55]. Experiments performed with seven reaction materials of VOCs showed that this process could convert VOCs into hydrogen-rich syngas used in solid oxidation fuel cells.

### 2.4. The Efficiency of ATR

Energy consumption (external heating value) and efficiency were the main indicators to evaluate autothermal reforming. Figure 7 shows the effect of different reactions on the total value via low heating values and indicates the comparison results of the actual power versus the theoretical power. According to Figure 7a, the efficiency value of the reforming process increased with the increase in reforming temperature showing that higher temperatures are beneficial for hydrogen production. However, the actual external heating values was different from the ideal external heating values because of large heat loss in the furnace. Figure 7b demonstrated that when O_2_/C > 0.2, the increase in oxygen-to-carbon led to a decrease in hydrogen production and the external heating value. The excess oxygen promoted the catalyst partial oxidation reaction to produce a large amount of heat, and then the required external heat was reduced. Figure 7c illustrates that the increase of steam-to-carbon may suppress catalytic partial oxidation reaction and promote steam reforming reaction increase hydrogen. However, the total external heating value increased rapidly. Considering the energy efficiency of the autothermal reforming holistically, the operating conditions in this study were chosen reasonably and could be effectively used as a reference for scaling up experiments with utilization in industrial applications.

## 3. Experiment and Analysis

### 3.1. Experiment Setup and Analysis

The experimental setup for autothermal reforming of toluene is shown in Figure 1. The experimental setup included the feeding, reaction, condensing, and gas analysis section. The nitrogen, hydrogen, and dry air gas were fed by MFC (sevenatar-D07) into the reactor. The deionized water and toluene were fed into the top of the quartz reaction tube by two horizontal pumps (szweico-2PB).

A commercial catalyst, with a loading of 16 wt% of NiO and K_2_O deposited on γ-Al_2_O_3_ was used for the following experiment. A quartz tube reactor was located in a furnace, the catalyst was placed in the center of the reactor inside two layers of silica wool. The external wall of the tube was connected with a temperature control device, and a movable thermocouple was inserted into the catalyst bed to detect the temperature of the catalyst bed during the reaction process.

The condensing section was consisted of a custom gas-liquid separator. The exhaust gas was fed in and gas-liquid separation was achieved. The gas products were generated by the gas-liquid separator and analyzed online by GC (Agilent column, Agilent Technologies Inc., Santa Clara, CA, USA: 3000 A, detector: FID) and TCD (Thermal Conductivity Detector, RISUN Technology, Shenzhen, China). Liquid samples were periodically taken via a sampling loop for HPLC analysis.

The reaction temperature was adjusted in the range of 550–700 °C. To ensure the accuracy of the experimental measurements, the horizontal flow pump and MFC used were calibrated using the standard curve method. The results presented in this study were obtained 30 min after the reaction system reached initial stability according to the stable components of the generated gas. The catalytic performance was evaluated by H_2_ yield and conversion. The following expressions were used to calculate the conversion of the feed (X), the hydrogen yield based on the gas phase product (Y_H2_), and carbon equilibrium (C%).
(9)$$Y_{H2}=\frac{n_{H_{2},out}}{n_{\left(feedstock\right)in,H_{2}}+n_{\left(H_{2}O\right)in,H_{2}}}\times 100\%$$
(10)$$X_{toluene}=\frac{n_{\left(feedstock\right),\;in}-n_{\left(feedstock\right),\;out}}{n_{\left(feedstock\right),\;in}}\times 100\%$$
(11)$$C-balance\left(\%\right)=\frac{C_{out}}{C_{in}}\times 100\%$$
where $n_{\left(feedstock\right)in,H_{2}}$ was the molar of H_2_ in the feedstock, $n_{\left(H_{2}O\right)in,H2}$ was the molar of H_2_ in the H_2_O, and $n_{H_{2},out}$ was the total amount of hydrogen produced in the reaction, $n_{\left(\mathrm{feedstock}\right),\;in}$ and $n_{\left(\mathrm{feedstock}\right),\;out}$ were molar ratios of feedstock at inlet and outlet. $C_{\mathrm{in}}$ and $C_{out}$ were the molar of feedstock at inlet and outlet.

### 3.2. Thermal Efficiency of Autothermal Reforming Process

In order to determine the advantages of autothermal reforming method and provide a scientific basis for a high-quality, high-yield, and high-efficiency reforming process, it was necessary to calculate the thermal efficiency. The thermal efficiency of the system was evaluated by the following Equation (12):(12)$$Effiency\%=\frac{LHV_{H_{2}}+LHV_{CH_{4}}}{LHV_{C_{7}H_{8}}+Heat_{absorbtion}}\times 100\%$$

In Equation (12), the $LHV_{H_{2}}$ was the total heating value of hydrogen, the $LHV_{CH_{4}}$ was the total heating value of CH_4_ and the where $LHV_{C_{7}H_{8}}$ was the total heat value of consumed fuel and$\;Heat_{absorbtion}$ was the total heat absorption [56].

According to Table 3, the ideal thermal efficiency of fuels reforming under different operation temperatures, oxygen-to-carbon ratios, and steam-to-carbon ratios can be calculated.

## 4. Conclusions

In this work, volatile organic compounds were reformed to hydrogen-rich by autothermal reforming at the optimal operating conditions (temperature = 700 °C, O_2_/C = 0.225, S/C = 1.31). The experimental results followed similar trends with simulation data in the autothermal reforming process, which indicates that this methodology could effectively guide the reaction process. Toluene was reformed to produce higher hydrogen yield and lower methane on NiO/K_2_O-γ-Al_2_O_3_ catalyst, which showed that the doping of potassium oxide can inhibit the methanation. We found that the temperature of the upper part of the catalyst bed is higher than the lower part, which demonstrated catalytic partial oxidation reacted at the top section of the catalyst bed and steam reformed at the lower section. The result proved that this system was self-sustaining and could effectively reduce energy consumption. Furthermore, actual VOCs (C_9_, C_10_, toluene, paraxylene, diesel, benzene, kerosene, and raffinate oil) feedstocks were also reformed to H*_2_*-rich syngas in ATR. Autothermal reforming of VOCs is a perspective and feasible process to offer fuel gas for SOFC.

## Data Availability

Data is contained within the article.

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
