# Peer review of "Autothermal Reforming of Volatile Organic Compounds to Hydrogen-Rich Gas"

_molecules, 2023, doi:10.3390/molecules28020752_

Round 1

Reviewer 1 Report

The manuscript entitled ‘Autothermal Reforming of Volatile Organic Compounds to Hydrogen-rich Gas’ presents a systematic study of VOCs conversion to hydrogen/syngas. Effect of operating conditions was investigated in terms of gases produced and bed temperature. An equilibrium model from ASPEN Plus was compared to that from the experiment. Overall, the manuscript was well organized and well written. The introductory part is sufficient in details with the background of VOCs reforming along with the objectives of this work. The methodology is well and sufficiently described. The results are well discussed according to the effect of varied operating conditions. The reviewer, however, has some comments for the authors’ consideration. 

  1. 1. Scheme 1 is suggested to be expanded to more detail especially for the reactor section. The position of temperature sensors installed for bed temperature measurement should be included and it would be useful when discussing temperature difference in the results section. Additionally, some dimensions, e.g., length of heating zone, catalyst bed height, reactor diameter may be added.  

  1. 2. The reviewer kindly suggests that the results from ‘Thermodynamic analysis’ may be discussed by comparing with some other works previously published. 

  1. 3. The discussion regarding temperature gradiences in catalyst bed is strongly suggested to be expanded. Probably temperature of the gas inlet, enthalpy of reaction, may be considered. 

Author Response

Dear Editors and Reviewers,

Thank you for your letter and for the reviewers’ comments concerning our manuscript entitled" Autothermal Reforming of Volatile Organic Compounds to Hydrogen-rich Gas" (ID: molecules-2133933). Those comments are all valuable and very helpful for revising and improving our paper, as well as the important guiding significance to our research. We feel great thanks for your professional review work on our manuscript. As you concerned, there are several problems that need to be addressed. According to your nice suggestions, we have studied the comments carefully and have made correction which we meet with approval. Revised portions (Track Changes) are marked in red in the manuscript. 

We tried our best to improve the manuscript and made some changes in revised manuscript which not influence the content and framework of manuscript. We appreciate for Editors and Reviewers’ warm work earnestly and hope the correction will meet with approval.

Special thanks to you for your professional and kindly comments. We are looking forward to hearing from you soon.

With best wishes,

Sincerely,

Chao Bian

Reviewer 2 Report

Author submitted the manuscript to report autothermal reforming VOCs to generate H2 rich gases. This topic is of interest to reduce VOCs and PM emissions and to overall reduce pollution. The manuscript have a decent combination of experimental and modeling data. The manuscript can be published after incorporating below changes.

Overall: Make sure that H2, C9, C10, etc are written correctly with the numbers in subscript. E.g., lines 124, 333, 374, 376, and wherever appropriate

Page 1, Introduction: Readers may not understand the unit of “135.5 Tg·C/Y”. You may expand it to 135.5 teragram Carbon/year or 135.5 x 109 kg Carbon/year

Page 6, Figure 1: What is your governing equation for the optimization to estimate T, O2/C, S/C ratio? Please mention the fundamental in the text or you may insert equation(s)

Page 6, Figure 1: Please use operating condition in the figure caption as it is not clear. For example, what is O2/C and S/C ratio in figure 1a, what is temperature (I think, 700C) and S/C ratio in figure 1b, and so on. Also, what is total flow rate of the reactants? Similar comment for other figures too.

Page 6: Put the space in front of 1.31 in the sentence “… equal to1.31.”

Page 8, Figure 3b: The cartoon on top-left shows Top and Back. Do you mean Bottom instead of Back – if yes, please correct it. Also, provide an arrow for the direction of gas flow. Same comment for figure 4b and 5b

Page 11, Figure 7: It is very difficult to read this figure as legend changes colors from a to b to c. Make sure you use same color and symbols for these three subplots. Also, put an arrow towards left for “Calculation Efficiency” for better readability

Author Response

(The authors gave the same response as above.)

Reviewer 3 Report

Dear Authors,

my comments are attached.

Best regards

Author Response

(The authors gave the same response as above.)
